# Single-Shot Phase Measuring Profilometry Based on Quaternary Grating Projection

**Chaozhi Yang, Yiping Cao * and Xiuzhang Huang**

College of Electronics and Information Engineering, Sichuan University, Chengdu 610065, China;
2018222050009@stu.scu.edu.cn (C.Y.); 2018222055222@stu.scu.edu.cn (X.H.)
* Correspondence: ypcao@scu.edu.cn; Tel.: +86-28-8546-3879

**Abstract:** In this paper, we propose a new single-shot three-dimensional (3D) measuring method based on quaternary grating projection. In traditional binary grating phase measuring profilometry (PMP), a multi-step or color fringe pattern are usually used to extract the sinusoidal fringes. In our proposed method, by using the DLP4500's 2-bit gray coding mode, the grayscale is quaternary. The three non-zero grayscales cyclically arranged in equal width, and the fourth grey value is 0 which is not encoded in the fringe pattern but represents the shadow information in the deformed pattern, where a quaternary grating is encoded. When the DLP4500 projects the quaternary grating onto the measured object, the charge coupled device (CCD) captures the corresponding deformed pattern synchronously. Three frames of binary deformed patterns with 1/3 duty cycle and a relative displacement of 1/3 period can be decomposed by the segmentation algorithm proposed in this paper. Three sinusoidal deformed patterns with a $2\pi/3$ shift-phase can be obtained by extracting the fundamental frequency of the three binary deformed patterns correspondingly, and the 3D shape of the object can be reconstructed by PMP. Experimental results show the effectiveness and feasibility of the proposed method. Because the DLP4500 only needs 2-bit coded grating for projection, the refresh rate of the projected grating is as high as 1428 Hz, which will have a broad application prospect in real time and fast online measurement.

**Keywords:** 3D shape and deformation measurement; phase measurement; single shot; real-time measurement

## 1. Introduction

In recent years, optical three-dimensional (3-D) shape measurement based on grating or fringe techniques has been widely used in machine vision, face recognition and product inspection with the good performance of high accuracy, high measurement efficiency, whole-field analysis and noncontact [1–7]. Due to the demand for fast real-time detection in the industrial field and face recognition, real-time 3D shape measurement technology has increasingly attracted attention [8–11]. Currently, the most widely applicable real-time 3D measurement technologies mainly include Fourier transform profilometry (FTP) [12,13], phase measuring profilometry (PMP) based on a high-speed projection system [14] and single-shot color fringe profilometry [15]. The Fourier transform profilometry (FTP) [16] proposed by Takeda et al. in 1983 uses a sinusoidal grating to obtain a single frame of a sinusoidal deformed pattern for real-time measurement, but its accuracy is limited by spatial frequency domain filtering. Although high-speed projection PMP has higher measurement accuracy [17], it is inefficient and the cost is expensive. Single-shot color profilometry [15] is performed by projecting a frame of color composite grating and the three phase-shifting sinusoidal fringes are encoded in the red (R), green (G), and blue (B) channels, respectively, to obtain the deformed patterns modulated by the object and then separate them from the R, G, and B components. The corresponding three frames of sinusoidal deformed patterns are separated to reconstruct the 3D shape of the object online. However, the color crosstalk among three color channels and the gray imbalance

in color fringe profiling impair the accuracy of the 3D shape reconstruction of the object to a certain extent.

Due to the DLP having a faster refresh rate at lower bit depth, many phase shift profilometry measurement based on binary gratings has been proposed to increase the detection speed of 3D detection. Su et al. proposed a technique using Ronchi grating to generate sinusoid grating fringe patterns by the method of defocusing binary structured ones [18]. However, it is not easy to calibrate the defocused projector and the depth range measurement of this method is small. Ekstand et al. demonstrated a nine or more step binary phase-shifting algorithm with a nearly focused projector, which has a significantly large depth range and highly accurate calibration [19], however, the accuracy and efficiency are limited by the wide period of binary grating and multi-phase-shifting steps. Li et al. compounded the sinusoid fringe into the binary color defocused projected fringe to reduce the nonlinear gamma [20]. However, this method has the limits of a specific defocusing device and the color crosstalk problem remains inevitable. Zuo et al. introduced the method of bi-frequency tripolar pulse-width modulation (TPWM) fringe projection and the sinusoid pulse-width modulation (SPWM) fringe projection. By optimizing the PWM properly, this method could eliminate the impact of the dominant undesired harmonics. However, the duty cycle is still 1/2 and the four-step phase-shifting algorithm is needed to reconstruct the object [21,22]. Su et al. proposed a one-shot projected fringe profilometry using a 2D fringe-encoded pattern, and the fringe pattern used for phase extraction can be directly utilized for unwrapping, but the accuracy of this method is limited by the FTP algorithm [23]. Cheng et al. [24] designed a contrast encoded pattern with a quaternary contrast-encoded scheme for phase-shifting techniques, however, three sinusoid quaternary contrast-encoded pattern are necessary to reconstruct the 3D profile of an object. Chen et al. [25] introduced a quantized phase-coding method to generate more than 36 different three-digit codes to achieve a high measurement accuracy, but the measurement speed was also limited by three fringe pattern projection. Heist et al. proposed an alternative measurement using an aperiodic sinusoidal fringe pattern with intensity profiles instead of phase-shifted purely sinusoidal fringes, where two cameras were used in the measurement system, thus it should be noticed that the array projector is unique, and the speed of the whole system is high [26]. Fu [27] et al. proposed a real-time 3D surface PMP based on color binary gratings, in which three monochromatic binary fringes occupy 1/3 period, respectively, and when arranged periodically, these binary fringes are encoded in red (R), green (G), and blue (B) channels, and the sinusoidal is directly extracted from the encoded binary grating without defocusing projection and using a proper filtering operation in spatial frequency domain. However, a color charge coupled device (CCD) camera must be adopted in this method, which will limit the overall reconstruction rate.

In this paper, we propose a new real-time 3-D shape measurement method based on single-shot quaternary grating projection.

## 2. Principle

### 2.1. The Quaternary Grating Design Principle

The traditional binary gratings in PMP are shown in Figure 1a,b. The duty cycle $\omega_0/T$ generally maintains 1/2 but can be 1/3 or else.

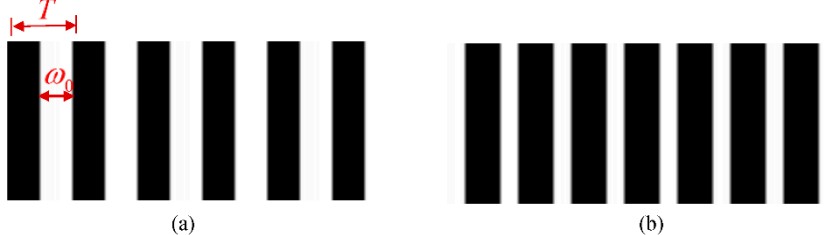

**Figure 1.** The binary gratings: (**a**) binary grating with 1/2 duty cycle; and (**b**) binary grating with 1/3 duty cycle.

The color binary fringe projection (CBF) proposed by Fu et al. [24]. Three monochromatic binary fringes occupy 1/3 period, respectively, and are arranged periodically. When the binary fringes are encoded in red (R), green (G), and blue (B) channels as shown in Figure 2, by separating the captured color fringe pattern into R, G, and B components, respectively, a real-time measurement can be achieved. However, to capture the fringe images, a color CCD must be adopted, and generally the frame capturing rate may be limited.



**Figure 2.** The features of the composite CBF and the segmentation.

The schematic diagram of a real-time 3D measuring method based on single-shot quaternary grating projection is shown in Figure 3. The quaternary grating is encoded and saved into the flash memory of the (Digital Light Processing) DLP. The DLP projects the quaternary grating with the period width of 9 pixels onto the reference plane and the object, and the project angle $\alpha$ is 27 degrees. At the same time, a high frame rate monochrome Charge-coupled Device (CCD) camera is capturing the corresponding fringe pattern synchronously on the same plane.

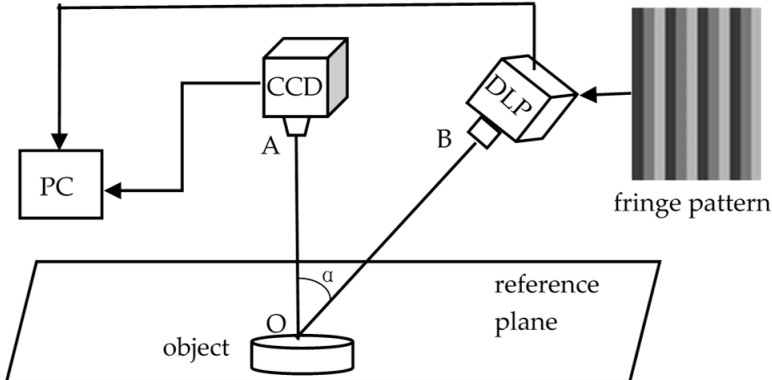

**Figure 3.** Online 3D measurement system based on quaternary grating projection.

The projected matrix of the quaternary grating is shown in Figure 4a. The three non-zero gray levels of 1, 2, and 3 are arranged and encoded in equal width in sequence. Figure 4b shows the cutaway view of the single-shot quaternary grating. When the quaternary grating is projected onto the reference plane and object, respectively, with DLP4500, by selecting the 2-bit coding mode, the correspondingly captured fringe pattern on the reference and the deformed pattern on the object are shown in Figure 4c,d. The shadow in the deformed pattern corresponds to the 0 grayscale in the quaternary grating projection as shown in Figure 4d, which is why the 0 grayscale is not encoded into the quaternary grating. Because the captured deformed pattern is of 256 grayscale, it is firstly segmented into a quaternary deformed pattern, and then decomposed into three binary deformed patterns, therefore the 3D shape of the object can be reconstructed by the binary grating projection PMP. Because the projected quaternary grating is encoded with only 2-bit depth with DLP-4500, the refresh rate can reach up to 1428 Hz. If the higher frame capturing rate a monochrome camera is used, it will have a broad application prospect in real-time and fast online measurement.

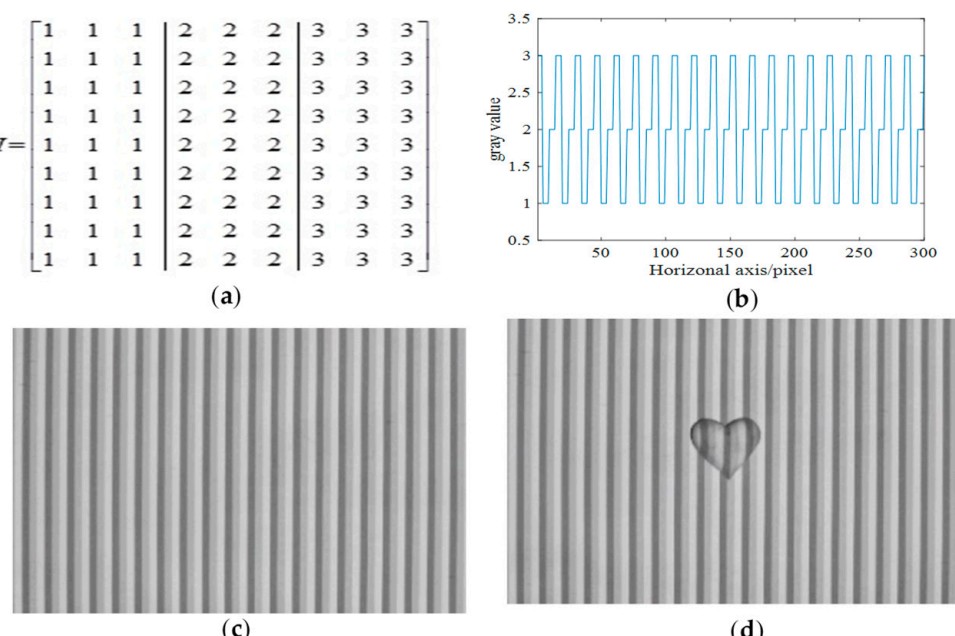

**Figure 4.** Quaternary grating encoding and capturing: (**a**) coding matrix; (**b**) the cutaway view of quaternary grating; (**c**) captured fringe pattern; and (**d**) captured deformed pattern.

### 2.2. Principle of Fringe Segmentation by Dynamic Threshold and Edge Detection

To decompose the essential three frames of binary deformed patterns with a duty cycle of 1/3 from the captured deformed pattern, a specific decomposition method should be carried out. The optical proximity effect (OPE) at the edges between each of the adjacent fringes with different grayscales in the projected quaternary grating must be considered. As shown in Figure 5a, there are several pixels of transition at these edges. Threshold segmentation will lead to incorrect segmentation on the boundary. Figure 5b shows that there are several pixels at the edge of the highest gray level and the lowest gray level that are classified as the intermediate gray levels. Therefore, we propose a fringe segmentation method based on fringe edge detection and region dynamic gray threshold detection.

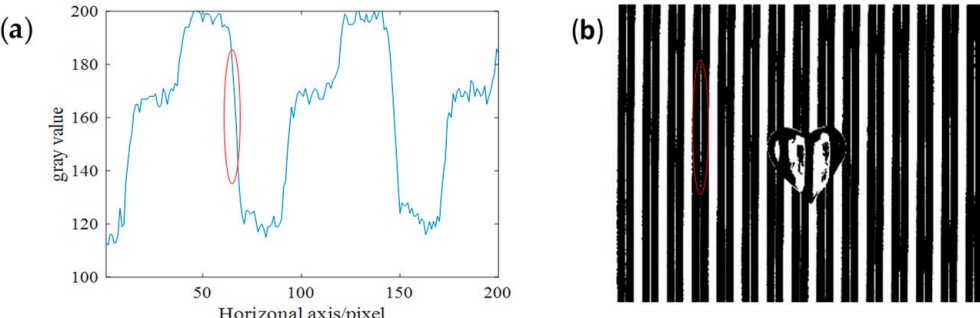

**Figure 5.** The optical proximity effect (OPE) on the boundary of the period of the deformed pattern: (**a**) the gray value distribution on the boundary of the period of the deformed pattern; and (**b**) the inaccurate segmentation of the deformed pattern.

The region dynamic gray threshold detection is based on the individual period process row by row. It is important to decompose the period into rows. Figure 6a, b shows the distribution of gray value and the differential values of the 513th row along the horizontal direction, so we can find the zero-crossing point easily, and the coordinates correspond to the fringe edges between the highest gray level and the lowest gray level, which are also the periodic boundaries.

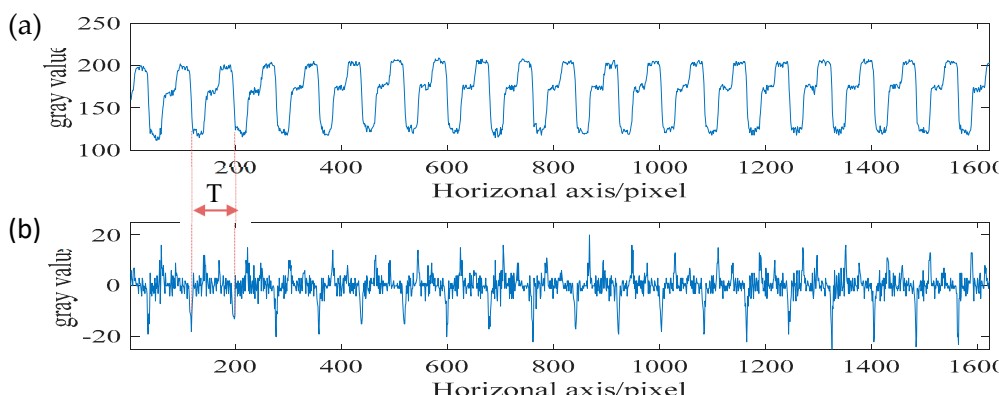

**Figure 6.** The gray value and differential value of the fringe patterns of the 513th row; (**a**) the distribution of the gray value; and (**b**) the distribution of differential value.

Therefore, each period in a row can be decomposed by the edge detection at each local minimum differential value, and the whole procedure of fringe segmentation is illustrated in Figure 7. Firstly, calculating the differential of the captured deformed pattern row by row; then, segmenting every period in the current row; and then, clustering the grayscales in every period. Because the cluster elements corresponding to the real grayscales among the different periods may be disparate, e.g., the cluster 1 represents the lowest grayscale in one period which is shadow information, but in another period, the amount of shadow pixels may be larger than the second grayscale and if the first cluster represents the second grayscale in this period, it is obviously a mistake. We must thus do a cluster element correspondence adjustment by read the real gray value of each cluster to make the correspondence between cluster elements and the real grayscale in each period appropriately. In this way, the captured deformed pattern is globally clustered to be another deformed pattern with only four grayscales (if the shadow exists) or three grayscales (if there is no shadow). If the cluster elements of this clustered deformed pattern are reassigned to 3, 2, 1 and 0 in the descending order of correspondence, where 0 represents the shadow, a quaternary deformed pattern demodulates successfully. While the three non-zero grayscales are picked out and pasted into the corresponding pixel positions on the three black images, respectively, after normalization, three binary deformed patterns with 1/3 duty cycle and 1/3 period displacement are decomposed. After that, a little bit of sporadic points may be wrong clustered, so an appropriate spatial filter is needed to optimize the three binary deformed patterns. It must be pointed out that this method can prevent incorrect segmentation, caused by the inhomogeneous illumination and the uneven surface reflectivity.

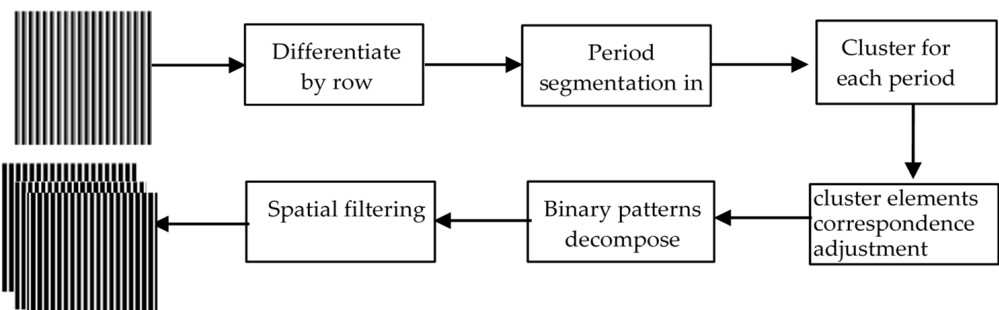

**Figure 7.** Fringe segment procedure.

### 2.3. The PMP Based on the Proposed Quaternary Grating Projection Principle

After obtaining three binary fringes with a duty cycle of $1/3$ and an interval of $1/3$ period, the binary grating PMP method could be used to reconstruct the 3D surface shape of the object. The binary deformed patterns can be expressed as

$$\begin{cases} I_1(x,y) = A_0 rect(\frac{x}{w_1}) * comb(\frac{x}{T_0}) \\ I_2(x,y) = A_0 rect(\frac{x}{w_1}) * comb(\frac{x-T_0/3}{T_0}) \\ I_3(x,y) = A_0 rect(\frac{x}{w_1}) * comb(\frac{x-2T_0/3}{T_0}) \end{cases} \tag{1}$$

where $I_1(x,y)$, $I_2(x,y)$, $I_3(x,y)$ represent the gray distribution of three nonzero grayscale deformed patterns. $A_0$ represents the background light intensity which is constantly played, w1 represents the width of a nonzero part in one period of the encoded binary fringe, and $T_0$ is the width of the period of the encoded binary fringe, and $*$ denotes the convolution operation.

The $I_1(x,y)$ is analyzed in detail to illustrate the principle of the proposed method. The Fourier spectrum $G_1(f_x, f_y)$ of $I_1(x,y)$ can be expressed as

$$G_1(f_x, f_y) = A_0 w_1 [\delta(f_x) + \sin c(\frac{w_1}{T_0})\delta(f_x + f_0) + \sin c(\frac{w_1}{T_0})\delta(f_x - f_0)] \tag{2}$$

The nearly unbroken sinusoidal deformed pattern $I_{1F}(x,y)$ can be extracted by Inver Fourier Transform (IFT) and represented as

$$I_{1F}(x,y) = A_0 P_{dc} T_0 + 2A_0 P_{dc} T_0 \sin c(P_{dc}) \cos(2\pi f_0 x) \tag{3}$$

The other two sinusoidal deformed patterns $I_{2F}(x,y)$ and $I_{3F}(x,y)$ can also be extracted from another two deformed patterns, respectively, which can be represented as

$$\begin{cases} I_{1F}(x,y) = A(x,y) + B(x,y) \cos[\frac{2\pi x}{T} + \varphi(x,y)] \\ I_{2F}(x,y) = A(x,y) + B(x,y) \cos[\frac{2\pi x}{T} + \varphi(x,y) + \frac{2\pi}{3}] \\ I_{3F}(x,y) = A(x,y) + B(x,y) \cos[\frac{2\pi x}{T} + \varphi(x,y + \frac{4\pi}{3})] \end{cases} \tag{4}$$

where T represents the period of the deformed patterns, $A(x,y)$ is simplified from $A_0 P_{dc} T$ and represents the background light intensity, $B(x,y)$ represent the contrast of the deformed pattern and simplified from $2A_0 P_{dc} T \sin c(P_{dc})$, $\varphi(x,y)$ is the phase introduced by the reference plane and it can be expressed as

$$\varphi(x,y) = \arctan\left\{ \frac{\sqrt{3}[I_{1F}(x,y) - I_{3F}(x,y)]}{2I_{2F}(x,y) - I_{1F}(x,y) - I_{3F}(x,y)} \right\} \tag{5}$$

Due to the $\varphi(x,y)$ is wrapped in $(-\pi, \pi]$ by the arctan function. The wrapped phase of the object should be unwrapped to $\phi_d(x,y)$, in the same way, the unwrapped phase of the reference plane can be acquired as $\phi_d(x,y)$, and the phase $\phi_d(x,y)$ caused by the height of the measured object can be represented as

$$\phi_h(x,y) = \phi_d(x,y) - \phi_r(x,y) \tag{6}$$

Thus, a phase-to-height mapping relationship is used to reconstruct the 3D shape of the measured object [28]:

$$\frac{1}{h(x,y)} = a(x,y) + b(x,y)\frac{1}{\phi_h(x,y)} + c(x,y)\frac{\phi_r(x,y)}{\phi_h(x,y)} \tag{7}$$

where $a(x,y)$, $b(x,y)$ and $c(x,y)$ are the system constants and can be calibrated by several planes with known heights.

### 3. The Experimental Results and Analysis

The measurement system of the proposed method is shown in Figure 8. It mainly consists of a DLP (Light Crafter 4500) and a CCD camera (HXC20). In the experiment, the three non-zero grayscale binary gratings share the same width in a period, which means that the duty cycle is 1/3. In our measurement system, the illumination of DLP4500 is LED with the illuminance of 150–500 lx the height measurement range is about 40 mm. The size of the figure captured by CCD is 1624 × 1236 pixels, which corresponds to the real field of a view size of 192.6 × 146.5 mm when the distance between the reference plane and the CCD camera is 700 mm, the lateral resolution of this system is about 0.118 mm. The width of the fringe period we project onto the object is 9 pixels, which expanded to 76 pixels in a deformed pattern, so that the height resolution could be calculated by the phase-to-height relationship which is about 0.013 mm. The measurement rate of this system is 300 frames per second.

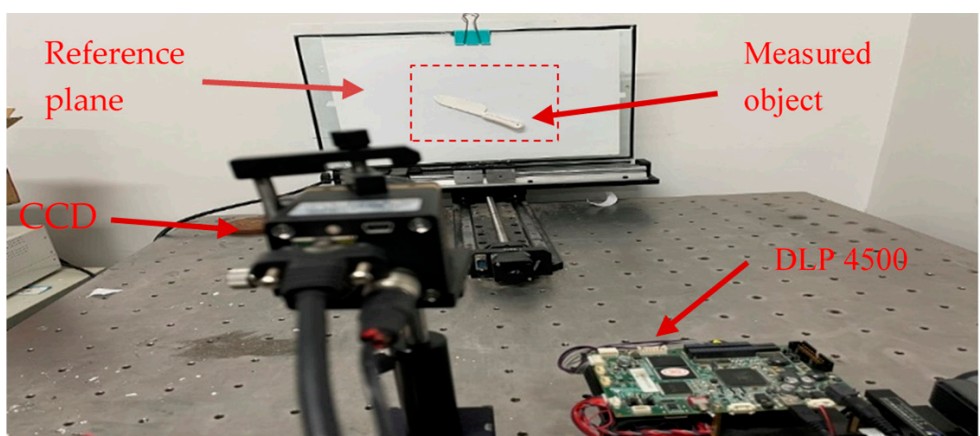

**Figure 8.** Schematic of experimental setup.

### 3.1. The Static Measurement Experiment

Multiple static objects are measured to verify the feasibility of the proposed method. Figure 9 shows one of the experimental results for a knife model measurement. Figure 9a,b show the knife model and its corresponding single-shot deformed pattern, and Figure 9c,d show the three frames binarized deformed patterns decomposed from the single-shot deformed pattern and the corresponding sinusoidal phase-shift deformed patterns, respectively. Figure 9e is the wrapped phase modulated by the height of the knife model, and Figure 9f is the 3D reconstruction result. It can be seen from Figure 9 that the proposed method can reconstruct objects accurately and verify the practicability of the proposed method. Due to the single-shot quaternary grating projected by DLP4500, the projection fringe refresh rate is as high as 1428 Hz, which will have broad application prospects in real-time and fast online measurement.

### 3.2. The Real-Time Online Measurement Experiment

A workpiece which size is 63 mm (D) × 51 mm (W) × 4.47 mm (H) on the conveyor is reconstructed to verify the real-time measurement capacity of the proposed method. Figure 10 shows the real-time measurement results during the workpiece transfer process. The workpiece to be reconstructed is transported at a speed of 20 mm/s. The CCD camera captures the quaternary deformed patterns of the workpiece at a refresh rate of 30 frames per second. The full size of the captured deformed pattern is 1624 × 1236 pixels, the size of the workpiece on pixels are 531 × 430, which means the real size measured by this method is 62.97 mm (D) × 51.12 mm (W). Figure 10a–c shows three frames captured deformed patterns in real time with an interval of 0.03 s. Figure 10d–f show the corresponding wrapped phase caused by the height of the workpiece, and Figure 10g–i show the 3D shape reconstruction results of the workpiece online. It can be seen that the quaternary

grating projection PMP can completely reconstruct the 3D shape of the workpiece online with good instantaneity. There are uneven shadows in the hollow area of the workpiece. With the quaternary grating projected onto the object, the grayscale of the shadow in the hollow corresponds to the zero-value grayscale in the captured deformed pattern, and the grayscale difference between the hollow shadow and the three non-zero fringe projected obliquely is considerable and the characteristics are obvious. Therefore, in the reconstruction results of the object, the outlines of the hollow holes of the workpiece have also been completely reconstructed. Figure 10j–l show the reconstruction result in a lower shutter speed, and the motion track of workpiece is more clearly visible.

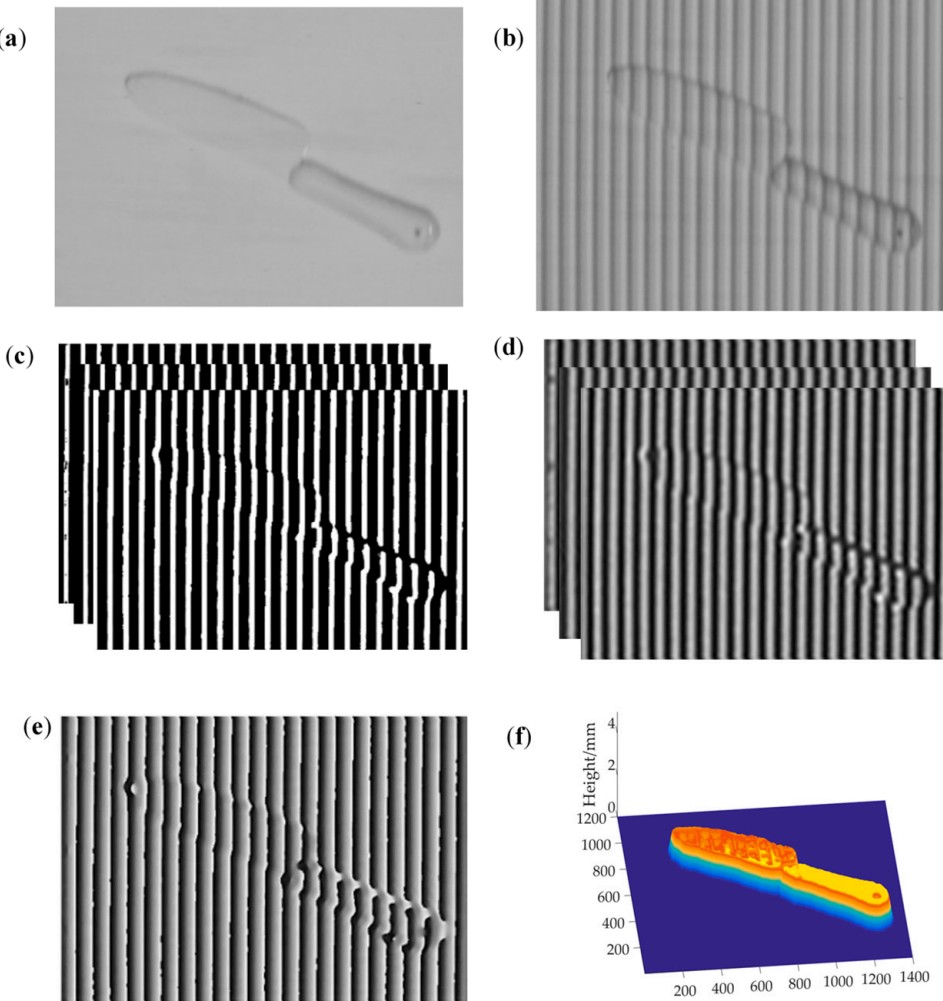

**Figure 9.** Knife measuring experiment: (**a**) the knife; (**b**) captured deformed patterns; (**c**) the binary deformed patterns; (**d**) extracted sinusoidal deformed patterns; (**e**) wrapped phase; and (**f**) reconstructed result.

### 3.3. Accuracy Analysis

To objectively verify the accuracy of the proposed method, the comparison experiment between FTP and the proposed method has been accomplished. Figure 11 shows the 550th column sectional view of the reconstructed 3D profile of the object. The blue dashed line is the result of FTP, and the red dashed line is the result of proposed method. It can be seen from Figure 11a that the result of the two fast real-time 3D shape measured technique are relatively close. However, from the details shown in Figure 11b, it can be clearly seen that on the boundary near the hollow, the outline of the hollow reconstructed by the proposed method is smooth and much closer to the real object than the result of the FTP algorithm.

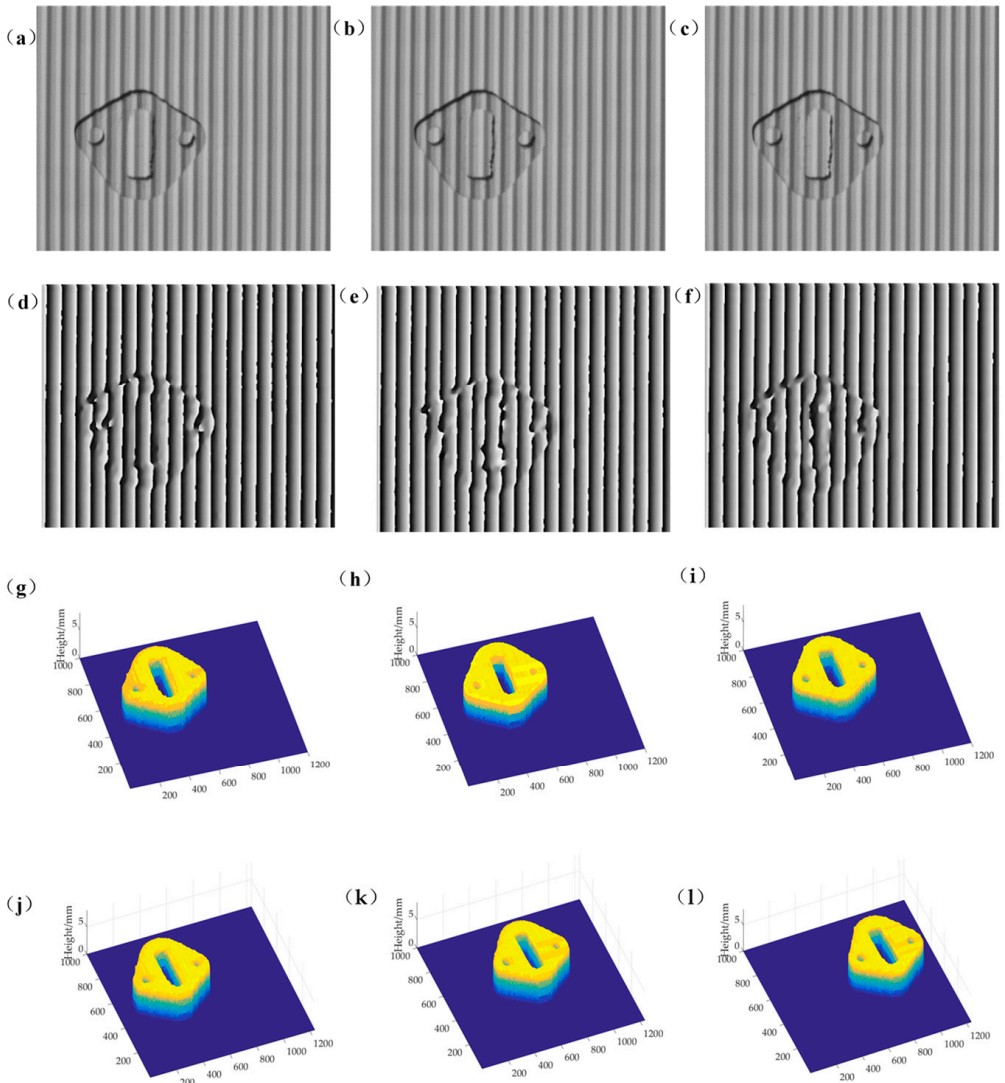

**Figure 10.** Real-time experiment for the workpiece: (**a–c**) the four grayscale deformed patterns captured online; (**d–f**) the corresponding wrapped phases; (**g–i**) the corresponding reconstructed results in high speed; and (**j–l**) the reconstructed results in lower shutter speed.

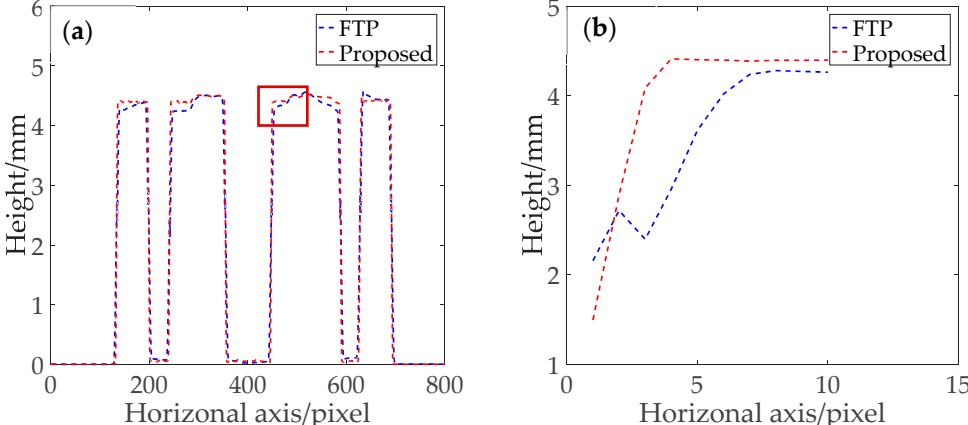

**Figure 11.** The 540th column data of the reconstructed object: (**a**) the sectional view of the 3D profile of the object; (**b**) the magnified view of the hollow boundary area of (**a**).

Moreover, in order to analyze the online measuring accuracy of the proposed method, the workpiece is measured at a different moving speed. As shown in Table 1, the Root Mean Squared Error (RMSE) of the height reconstructed by the proposed method remains between 0.034 and 0.048 mm. The height RMSE is very small compared to the known height of 4.47 mm. Thus, the quaternary grating projection PMP method is optional.

**Table 1.** The RMES in different moving speed of workpiece.

| Moving Speed (mm/s) | 5 | 10 | 20 | 30 |
|:---:|:---:|:---:|:---:|:---:|
| RMSE (mm) | 0.034 | 0.037 | 0.045 | 0.048 |

### 4. Conclusions

We proposed a real-time PMP method based on quaternary grating projection with a quaternary grating which is cyclically encoded with three one, two, three grayscale fringes and let the 0 grayscale correspond to the shadow. The quaternary grating is projected by DLP4500 and only one deformed pattern needs to be captured. After the specific segmentation, clustering and quaternization, three-frame interval 1/3 period equal width binarized deformed patterns can be decomposed from the captured deformed pattern. Therefore, the 3D shape of the measured object can be reconstructed by using these three binarized deformed patterns. The physical model is systematically derived. The experimental results show the effectiveness and practicability of the proposed real-time PMP method. By using DLP4500 with 2-bit depth coding mode, the quaternary grating achieves 1428 Hz high-speed projection A monochrome CCD camera is used for image capturing in a corresponding high refresh rate mode. The single-shot feature is fully capable of real-time online measurement. Compared with the traditional unequal duty cycle binary grating PMP method, it has the advantages of higher detection speed and good real-time performance. Compared with the real-time PMP method of color-coded binary grating, there is no color crosstalk and the efficiency of CCD is higher and the cost may be lower. Therefore, it has broad application prospects in the field of industrial fast real-time online 3D inspection.

**Author Contributions:** Conceptualization, C.Y. and Y.C.; methodology, C.Y.; validation, Y.C., X.H.; investigation, Y.C. and X.H.; resources, C.Y.; writing—original draft preparation, Y.C.; writing—review and editing, C.Y. and Y.C.; visualization, Y.C.; project administration, C.Y. All authors have read and agreed to the published version of the manuscript.

**Funding:** This research was funded by Special Grand National Project of China, 2009ZX02204-008.

**Institutional Review Board Statement:** Not applicable.

**Informed Consent Statement:** Not applicable.

**Data Availability Statement:** Data is contained within the article or supplementary material. The data presented in this study are available in Single-Shot Phase Measuring Profilometry Based on Quaternary Grating Projection.

**Conflicts of Interest:** The authors declare no conflict of interest.

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
