# Peer review of "Single-Shot Phase Measuring Profilometry Based on Quaternary Grating Projection"

_applsci, doi:10.3390/app11062536_

Round 1

Reviewer 1 Report

The paper describes a single-shot fringe projection method to extract 3D images using a 4-level grayscale fringe pattern projected using a DLP device that is capable of very high speed image acquisition using a phase shifting algorithm.  The method is well described and referenced but I have the following observations and questions:

On Line 59, what does the "T" stand for in TPWM fringe projection?

In Figure 3, what is the angle of the fringe projection?

What is the height resolution, range and depth of field without phase ambiguity, and what determines both?  This is important for any 3D method.  What is the lateral resolution and what determines it?  The width of the projected fringes?  The camera resolution?  Something else?

For the real-time measurement to highlight the movement you should zoom in on the object so the reader can better see the edge of the object with respect to the fringes to emphasize that the object has in fact moved for each frame.  I can see it in the full-field image but zooming in would make it more apparent.

Fig. 9:  Please change the scale in the z (height) dimensions to better show the 3D object.  20 mm max height is not appropriate for a relatively shallow object.

Figure 10:  Please change height scale to better show detail.  Also, if you can really see to the bottom of the holes without shadowing, that is significant, although with angled projection I don’t see how.  In Line 210 you write:  "Therefore, in the reconstruction results of the object, the hollow holes of the workpiece have also been completely reconstructed."  I assume you mean the aperture or outline of the hole, not the hole in its entirety of its depth to the bottom of the hole.  This should be shadowed even in this method.  

Author Response

Thank you very much for your excellent editing work of our manuscript entitled .My manuscript has been revised according to your comments. Please see some details in the attached file.

Reviewer 2 Report

It general it is a good paper. The major remarks, which require some adjustment are:

Literature: for the overview of the field, some groups around the world should be added who have been working on this field for years. Details see attached file.

Experiments:

The results should be quantified better. Saying that a reconstruction is good, is insufficient. Also, the experimental parameters like camera use, projection angle, etc. should be described in more detail (if needed in an appendix).

Please see some details in the attached file.

Author Response

(The authors gave the same response as above.)

Reviewer 3 Report

The manusctipt is prepared in proper fashion, no major issues appeared.

Yet, I suggest that authors proving the novelty of the work, refer to following works:

- Nai-Jen Cheng, Sih-Yue Chen, and Wei-Hung Su "Profile measurements using contrast-encoded pattern projections", Proc. SPIE 11123, Photonic Fiber and Crystal Devices: Advances in Materials and Innovations in Device Applications XIII, 111230U (9 September 2019); https://doi.org/10.1117/12.2530714 
- Wei-Hung Su and Sih-Yue Chen "One shot projected fringe profilometry using a 2D fringe-encoded pattern", Proc. SPIE 10755, Photonic Fiber and Crystal Devices: Advances in Materials and Innovations in Device Applications XII, 1075511 (4 September 2018); https://doi.org/10.1117/12.2323809 
- Wang Yuwei, Chen Xiangcheng, Wang Yajun. Modified dual-frequency geometric constraint fringe projection for 3D shape measurement[J]. Infrared and Laser Engineering, 2020, 49(6): 20200049. doi: 10.3788/IRLA20200049

Also, in order to reveal the resolution, accuracy and repeatability of the method, some estimation/ quantitative evaluation is necessary. Also, while the experimental results of fast reconstruction is presented, the comparison of following process outcomes should be provided, as the input to abovementioned parameters.

It would be also valuable to address the processing unit efficiency requirements.

Finally, some language corrections and editing work is still necessary (i.e. lines: 38, 71, 76, 163, 167, 168, 174.

Author Response

(The authors gave the same response as above.)

Round 2

Reviewer 2 Report

The provided changes are good and well placed. Adding the actual sizes of the objects, dimensions of the setup and the accuracy analysis improve the paper considerably.

Reviewer 3 Report

Still some minor language and editing issues can be found (in particular in new parts of the manuscript).